# MR. Estimator, a toolbox to determine intrinsic timescales from subsampled spiking activity

F. P. Spitzner[1]*, J. Dehning[1], J. Wilting[1], A. Hagemann[1], J. P. Neto[1], J. Zierenberg[1], V. Priesemann[1,2]*

1 Max-Planck-Institute for Dynamics and Self-Organization, Göttingen, Germany, 2 Bernstein-Center for Computational Neuroscience (BCCN) Göttingen, Göttingen, Germany

* paul.spitzner@ds.mpg.de (EPS); viola.priesemann@ds.mpg.de (VP)

**Data Availability Statement:** Referenced scripts are available at https://github.com/Priesemann-Group/mrestimator/blob/v0.1.7/examples/paper

## Abstract

Here we present our Python toolbox "MR. Estimator" to reliably estimate the intrinsic time-scale from electrophysiologal recordings of heavily subsampled systems. Originally intended for the analysis of time series from neuronal spiking activity, our toolbox is applicable to a wide range of systems where subsampling—the difficulty to observe the whole system in full detail—limits our capability to record. Applications range from epidemic spreading to any system that can be represented by an autoregressive process. In the context of neuroscience, the intrinsic timescale can be thought of as the duration over which any perturbation reverberates within the network; it has been used as a key observable to investigate a functional hierarchy across the primate cortex and serves as a measure of working memory. It is also a proxy for the distance to criticality and quantifies a system's dynamic working point.

## 1 Introduction

Recent discoveries in the field of computational neuroscience suggest a major role of the so-called intrinsic timescale for functional brain dynamics [1–8]. Intuitively, the intrinsic time-scale characterizes the decay time of an exponentially decaying autocorrelation function (in this work and in many contexts it is synonymous to the autocorrelation time). Exponentially decaying correlations are commonly found in recurrent networks (see e.g. Refs. [5, 9]), where the intrinsic timescale can be related to information storage and transfer [10–12]. More importantly, such decaying autocorrelations are also found in the network-spiking-dynamics recorded in the brain: Here, the intrinsic timescale serves as a measure to quantify working memory [3, 4] and unravels a temporal hierarchy of processing in primates [1, 2].

Although autocorrelations and the intrinsic timescale can be derived from single neuron activity, they characterize the dynamics within the whole recurrent network. The single neuron basically serves as a readout for the local network activity. One can consider spiking activity in a recurrent network as a branching or spreading process, where each presynaptic spike triggers on average a certain number $m$ of postsynaptic spikes [13–15]. Such a spreading process

Simulation data are available at https://doi.org/10.12751/g-node.licm4y.

**Funding:** FPS and JD were funded by the Volkswagen Foundation through the SMARTSTART Joint Training Program Computational Neuroscience. JZ is supported by the Joachim Herz Stiftung. All authors acknowledge funding by the Max Planck Society. The funders had no role in study design, data collection and analysis, decision to publish, or preparation of the manuscript.

**Competing interests:** The authors have declared that no competing interests exist.

typically features an exponentially decaying autocorrelation function, and the associated time constant is in principle accessible from the activity of each unit. However, approaching the single-unit level, the magnitude of the autocorrelation function can be much smaller than expected, and can be disguised by noise.

In experiments we approach this level: we typically sample only a small part of the system, sometimes only a single or a dozen of units. This subsampling problem is especially problematic in neuroscience, where even the most advanced electrode measurements can record at most a few thousand out of the billions of neurons in the brain [16, 17]. However, we recently showed that this spatial subsampling only biases the magnitude of the autocorrelation function (of autoregressive processes) and that—despite the bias—the associated intrinsic timescale can still be inferred by using multi-step regression (MR). Because the intrinsic timescale inferred by MR is invariant to spatial subsampling, one can infer it even when recording only a small set of units [5].

Here, we present our Python toolbox "MR. Estimator" that implements MR to estimate the intrinsic timescale of spiking activity, even for heavily subsampled systems. Since our method is based on spreading processes in complex systems, it is applicable beyond neuroscience, e.g. in epidemiology or social sciences such as the timescale of epidemic spreading (from subsampled infection counts) [5] or the timescale of opinion spreading (from subsampled social networks) [18].

The main advantage of using our toolbox over a custom implementation to determine intrinsic timescales is that it provides a consistent way that can now be adopted across studies. It supports trial structures and we demonstrate how multiple trials can be combined to compensate for short individual trials. Lastly, the toolbox calculates confidence intervals by default, when a trial structure is provided.

In the following, we discuss how to apply the toolbox using a code example (Sec. 2). We then briefly focus on the neuroscience context (including a real-life example, Sec. 3) before we derive the MR estimator and discuss technical details such as the impact of short trials (Sec. 4). While of general interest, this section is not required for a general understanding of the toolbox. In the discussion (Sec. 5), we present selected examples where intrinsic timescales play an important role. Lastly, an overview of parameters and toolbox functions is given in Tables 1 and 2 at the end of the document.

## 2 Workflow

To illustrate a typical workflow, we now discuss an example script that generates an overview panel of results, as depicted in Fig 1. The discussed script and other examples are provided online [19].

In the example, we generate a time series from a branching process with a known intrinsic timescale (Fig 1A). At the discrete time steps $\Delta t$ of such a branching process, every active unit activates a random number of units (on average $m$ units) for the next time step. As this principle holds for any unit, activity can spread like a cascade or avalanche over the system. Taking the perspective of the entire system, the current activity $A_t$ (or number of active units) depends on the previous activity and the branching parameter $m$. Then, the branching parameter is directly linked to the intrinsic timescale $\tau = -\Delta t/\ln(m)$: As $m$ becomes closer one, $\tau$ grows to infinity (for the mathematical background, see Sec. 4). Because $\tau$ corresponds to the decay time of the autocorrelation function (Fig 1C), a larger $\tau$ will cause a slower decay.

With this motivation in mind, it is the main task of the toolbox to determine the *correlation coefficients* $r_k$—that describe the autocorrelation function of the data—and to *fit* an analytic autocorrelation function to the determined $r_k$—which then yields the intrinsic time scale. In

**Table 1. List of the most common parameters and functions where they are used.** For a full list of each function's possible arguments, please refer to the online documentation [43].

| Symbol | Parameter description | Function | Example argument |
|---|---|---|---|
| $k$ | Discrete time steps of correlation coefficients (shift between original and delayed time series) | `full_analysis()` | `kmax = 1000` |
| | | `coefficients()` | `steps=(1, 1000)` |
| | | `fit()` | `steps=(1, 1000)` |
| | Unit of discrete time steps | `full_analysis()` | `dtunit='ms'` |
| | | `coefficients()` | `dtunit='ms'` |
| | | `fit()` | `dtunit='ms'` |
| $\Delta t$ | Size of the discrete time steps in `dtunits` | `full_analysis()` | `dt = 4` |
| | | `coefficients()` | `dt = 4` |
| | | `fit()` | `dt = 4` |
| $r_k$ | Correlation coefficients | `fit()` | `data` |
| | Method for calculating $r_k$ | `full_analysis()` | `coefficientmethod='sm'` |
| | | `coefficients()` | `method='ts'` |
| | Selecting Fitfunctions: | `full_analysis()` | `fitfuncs=['exp', 'offset', 'complex']` |
| | | `fit()` | `fitfunc='exp'` |
| $\alpha$ | Subsampling fraction | `simulate_subsampling()` | `prob` |
| | | `simulate_branching()` | `subp` |
| $\langle A_t \rangle$ | Activity (e.g. of a branching process) | `simulate_branching()` | `a = 1000` |
| $m$ | Branching parameter | `simulate_branching()` | `m = 0.98` |
| $h$ | External input | `simulate_branching()` | `h = 100` |
| | Bootstrapping: number of samples, rng seed | `full_analysis()` | `numboot = 100, seed = 101` |
| | | `coefficients()` | `numboot = 100, seed = 102` |
| | | `fit()` | `numboot = 100, seed = 103` |

the example, we determined $r_k$ with the toolbox's default settings (Fig 1C) and we fitted two alternative exponentially decaying functions to determine the intrinsic timescale (a plain exponential and an exponential that is shifted by an offset). The toolbox returns estimates and 75% confidence intervals for the branching parameter and the intrinsic timescale (Fig 1D); the estimates match the known values $m = 0.98$ and $\tau \approx 49.5$ that were used in the example. To demonstrate the effect of subsampling in the example, we recorded only 5% of the occurring events of the branching process.

**Listing 1**. Example script (Python) that creates artificial data from a branching process and performs the multistep regression. An example to import experimental data is available online, along with detailed documentation explaining all function arguments [19].

**Table 2. The (lengthy) descriptions of fit-functions and coefficient-methods can be abbreviated.**

| Full name | Abbreviation |
|---|---|
| `'trialseparated'` | `'ts'` |
| `'stationarymean'` | `'sm'` |
| `'exponential'` | `'e'`, `'exp'` |
| `'exponential_offset'` | `'eo'`, `'exp_offset'`, `'exp_off'` |
| `'complex'` | `'c'`, `'cplx'` |

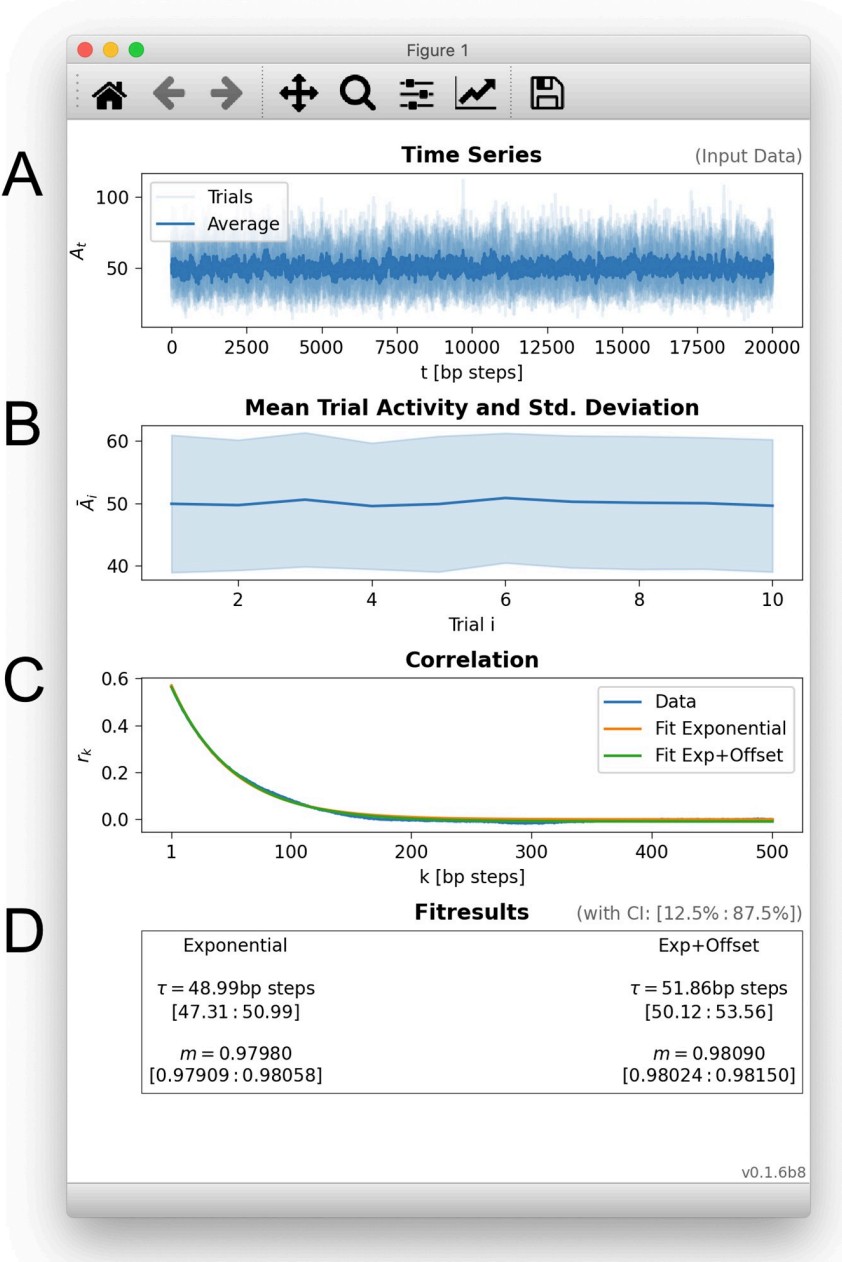

**Fig 1. The toolbox provides a `full_analysis()` function that performs all required steps and produces an overview panel. A**: Time series of the input data, here the activity $A_t$ of ten trials of a branching process with $m = 0.98$ and $\tau = \Delta t/\ln(m) \approx 49.5$ steps ($\Delta t$ is the step size of the branching process). **B**: Mean activity and standard deviation of activity for each trial. This display can reveal systematic drifts or changes across trials. **C**: Correlation coefficients $r_k$ are determined from the input data, and exponentially decaying autocorrelation functions are fitted to the $r_k$. Several alternative fit functions can be chosen. **D**: The decay time of the autocorrelation function corresponds to the intrinsic timescale $\tau$, and allows to infer the corresponding branching parameter $m$. The shown fit results contain confidence intervals in square brackets (75% by default).

```
# load the toolbox
import mrestimator as mre
# enable matplotlib interactive mode so
# figures are shown automatically
mre.plt.ion ()
# 1. -------------------------------#
# example data from branching process
bp = mre. simulate_branching (m = 0.98, a = 1000,
subp = 0.05, length = 20000, numtrials = 10, seed = 43771)
# make sure the data has the right format
src = mre.input_handler (bp)
# 2. -------------------------------#
# calculate autocorrelation coefficients,
# embed information about the time steps
rks = mre.coefficients (src, steps = (1, 500), dt = 1, dtu-
nit = 'bp steps', method = 'trialseparated')
# 3. -------------------------------#
# fit an autocorrelation function, here
# exponential (without and with offset)
fit1 = mre.fit(rks, fitfunc = 'exp')
fit2 = mre.fit(rks, fitfunc = 'exp_offset')
# 4. -------------------------------#
# create an output handler instance
out = mre.OutputHandler ([rks, fit1, fit2])
# save to disk
out.save ('~/mre_example/result')
# 5. -------------------------------#
# gives same output with other file title
out2 = mre.full_analysis (data = bp, dt = 1, kmax = 500,
dtunit = 'bp steps', coefficientmethod = 'trialseparated',
fitfuncs = ['exp', 'exp_offset'], targetdir = '~/mre_exam-
ple/')
```

**1**. **Prepare data**: After the toolbox is loaded, the input data needs to be in the right format: a 2D NumPy array [20–22]. To support a trial structure, the first index of the array corresponds to the trial (even when there is only one trial), the second index corresponds to the time (in fixed time steps). All trials need to have the same length.

We provide an optional `input_handler()` that tries to guess the passed format and convert it automatically. For instance, it can check and convert data that is already loaded (as shown in Listing 1) or load files from disk, when a file path is provided.

**2**. **Multiple regressions**: Once the data is in the right format, multiple linear regressions are initiated by calling `coefficients()` (see Sec. 4.3 for more details). The function performs linear regressions between the original time series (`src`), and the same time series after it was shifted by $k$ time steps. It returns the slopes found by the regression—we call them correlation coefficients $r_k$ (`rks`). Here, we specify to calculate the correlation coefficients for steps $1 \leq k \leq 500$. In Listing 1, the linear regression is performed for each trial separately. To obtain a joint estimate across all trials, the estimated $r_k$ are averaged (`trialseparated` method). Confidence intervals are calculated using bootstrapping.

Please note that (independent from subsampling) the linear regression can be biased due to short trials [23, 24]. In case of stationary activity across trials, the issue can be circumvented by using the `stationarymean` method (see Sec. 4.3 and Fig 5).

**3**. **Fit the autocorrelation function**: Next, we fit the correlation coefficients using a desired function (`fitfunc`). In order to estimate the intrinsic timescale, this function needs to decay exponentially. Motivated by recent experimental studies [1], the default function is `exponential_offset` (other options include an `exponential` and a `complex` fit with empirical corrections).

**4**. **Visualize and store results**: Multiple correlation coefficients and fits can be exported using an instance of `OutputHandler`. The `save()` function not only exports a plot but also a text file containing the full information that is required to reproduce it.

**5**. **Wrapping up**: For convenience, the `full_analysis()` function performs all steps with default parameters and displays an overview panel as shown in Fig 1.

## 3 Interpretation in a neuroscience context

Timescales of neural dynamics have been analyzed in various contexts and can be interpreted as reward memory [25] or as temporal receptive windows [26]. Here, however, we focus on the timescale of the decay of the autocorrelation function [1], which is thought to be related to the duration of integration in local circuits [2] or to working memory [3, 4]. As such, the intrinsic timescale represents a measure of how long information is kept (or can be integrated) in a local circuit; it ranges between 50 to 500 ms and this diversity of timescales is believed to arise from differences in local connectivity [27, 28].

In the brain, the autocorrelation function is not only determined by the intrinsic timescale. If the spiking activity is dominated by a single timescale $\tau$, the autocorrelation is expected to decay exponentially (see Sec. 4): $C(k) \propto \exp\left(-\frac{k}{\tau}\right)$. However, often the autocorrelation is more complex, which we take into account and provide a `complex` fit function, based on an empirical analysis of autocorrelation functions by König [29]:

$$C(k) = D e^{-\frac{k}{\tau_{\mathrm{exp}}}} + E e^{-\left(\frac{k}{\tau_{\mathrm{osc}}}\right)^{\gamma}} \cos\left(2\pi\nu k\right) + F e^{-\left(\frac{k}{\tau_{\mathrm{gauss}}}\right)^{2}} + O \ . \tag{1}$$

In addition to the exponential decay, the `complex` fit function features three terms that account for:

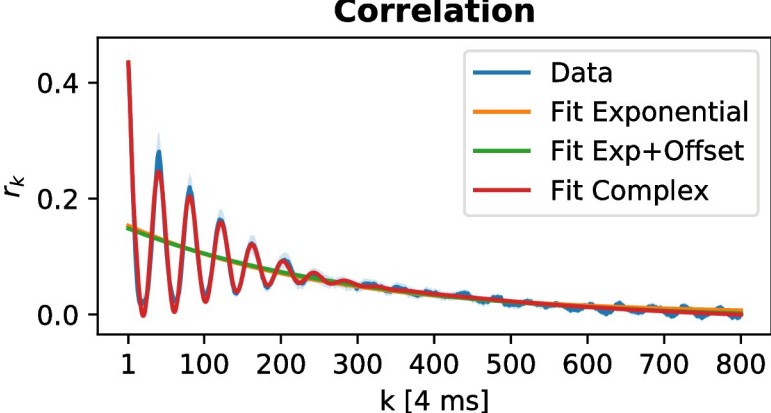

**Fig 2. Example analysis of spiking activity from rat hippocampus during an open field task that demonstrates the usage of the `complex` fit function.** A short example code that analyzes the data [30] and produces this figure is listed in appendix A.

- Neural oscillations, reflected as an exponentially decaying cosine term: $E e^{-\left(\frac{k}{\tau_{\text{osc}}}\right)^{\gamma}} \cos(2\pi v k)$.

- Short term dynamics of a neuron with a refractory period, reflected as a Gaussian decay:

$$F e^{-\left(\frac{k}{\tau_{\text{gauss}}}\right)^{2}}.$$

- An offset $O$ which arises due to the small non-stationarities of the recordings on timescales longer than a few seconds.

To illustrate the usefulness of the `complex` fit function, we analyze an openly available dataset of spiking activity in rat hippocampus [30]. We find an intrinsic timescale of around 1.5 seconds (which is similar to the timescales found in rat cortex [31]). One challenging characteristic of this dataset are theta oscillations (5–10 Hz) in the population activity, which carry over to the autocorrelation function. Because the `complex` fit function features an oscillatory term, it can capture these oscillations, and still yield a solid estimate of the autocorrelation time. (Fits from functions without the oscillatory term will deviate from the data and lead to biased estimates.) Additionally, by including this term into the fit, we also obtain an estimate of the oscillation frequency: In the shown example (Fig 2), we find $v = 6.1$ Hz, which is well in the range of theta oscillations. This shows that our toolbox can deal with complex neuronal dynamics of single-cell activity.

# 4 Technical details

## 4.1 Derivation of the multi-step regression estimator for autoregressive processes

The statistical properties of activity propagation in networks can be approximated by a stochastic process with an autoregressive representation [15, 18, 32], at least to leading order [14]. We will use this framework of autoregressive processes to derive the multi-step regression estimator and show that it is invariant under subsampling [5].

Here, we consider the class of stochastic processes with an autoregressive representation of first order. This process combines a stochastic, internal generation of activity with a stochastic, external input. The internal generation on average yields $m$ new events per event, where $m$ is

called the branching parameter (using the terminology of the driven branching process) [33–35]. The external input is assumed to be an uncorrelated Poisson process with rate $h$ (a generalization to non-stationary input can be found in Ref. [36]). For discrete time steps $\Delta t$, we denote the number of active units at time $t$ with $A_t$ and obtain the autoregressive representation

$$\langle A_{t+1}|A_t\rangle = mA_t + h\Delta t \,, \tag{2}$$

where $\langle\cdot\rangle$ denotes the expectation value. This autoregressive representation is the basis of our subsampling invariant method and makes it applicable to the full class of first-order autoregressive processes. From Eq (2), we can also see that one could determine $m$ from a time series of a system's activity by using linear regression. The *linear regression* estimate of $m$ is

$$m_{\text{lr}} = \frac{\text{Cov}[A_{t+1}, A_t]}{\text{Var}[A_t]} = \frac{\sum_{t=1}^{T-1}(A_{t+1} - \langle A_{t+1}\rangle)(A_t - \langle A_t\rangle)}{\sum_{t=1}^{T-1}(A_t - \langle A_t\rangle)^2} \,. \tag{3}$$

This well established approach [5, 33, 37, 38] only considers the pairs of activity that are separated by one time step—it measures the slope of the line that best describes the point cloud $(A_{t+1}, A_t)$. Instead, the *multi-step regression* (MR) estimator considers all the pairs of activity separated by increasing time differences $k$—it estimates multiple regression slopes.

Analogous to the case of $k = 1$ in Eq (3), we define the *correlation coefficients $r_k$* as the slope of the line that best describes the point cloud $(A_{t+k}, A_t)$

$$r_k \frac{\text{Cov}[A_{t+k}, A_t]}{\text{Var}[A_t]} = \frac{\langle(A_{t+k} - \langle A_{t+k}\rangle)(A_t - \langle A_t\rangle)\rangle}{\langle A_t^2\rangle - \langle A_t\rangle^2} \,. \tag{4}$$

For an autoregressive process that is fully sampled, these correlation coefficients become $r_k = m^k$. To show this, we first generalize Eq (2) using the geometric series (cf. Ref. [5, 36])

$$\langle A_{t+k}|A_t\rangle = m^k A_t + h\Delta t \frac{1 - m^k}{1 - m} \,. \tag{5}$$

We then use the law of total expectation to obtain $\langle A_{t+k} A_t\rangle = \langle\langle A_{t+k}|A_t\rangle A_t\rangle$ and $\langle A_{t+k}\rangle = \langle\langle A_{t+k}|A_t\rangle\rangle$. This allows us to rewrite the covariance:

$$\text{Cov}[A_{t+k}, A_t] = \langle A_{t+k} A_t\rangle - \langle A_{t+k}\rangle\langle A_t\rangle \tag{6}$$

$$= \langle\langle A_{t+k}|A_t\rangle A_t\rangle - \langle\langle A_{t+k}|A_t\rangle\rangle\langle A_t\rangle \tag{7}$$

$$= m^k\langle A_t^2\rangle + h\Delta t \frac{1 - m^k}{1 - m}\langle A_t\rangle - m^k\langle A_t\rangle^2 - h\Delta t \frac{1 - m^k}{1 - m}\langle A_t\rangle \tag{8}$$

$$= m^k(\langle A_t^2\rangle - \langle A_t\rangle^2) = m^k \text{Var}[A_t] \,. \tag{9}$$

When we insert this result into Eq (4), we find that the correlation coefficients are related to the branching parameter as $r_k = m^k$, which enables the toolbox to detect the branching parameter from recordings of processes that are subcritical ($m < 1$), critical ($m = 1$) or supercritical ($m > 1$).

In the special case of stationary activity, where $\langle A_t\rangle = \langle A_{t+k}\rangle$, the correlation coefficients can be further related to an autocorrelation time. In this case, the correlation coefficients, Eq (4),

match the correlation function

$$r_k = \frac{\langle A_{t+k} A_t \rangle - \langle A_t \rangle^2}{\langle A_t^2 \rangle - \langle A_t \rangle^2} = C(A_{t+k}, A_t). \tag{10}$$

Note that we here consider the definition of the autocorrelation function normalized to the time-independent variance (other definitions are also common, e.g. a time-dependent Pearson correlation coefficient Cov $[A_{t+k}, A_t]$ / Std $[A_t]$ Std $[A_{t+k}]$). For stationary autoregressive processes, the correlation function decays exponentially and we can introduce an autocorrelation time $\tau$

$$C(A_{t+k}, A_t) = e^{(-k\,\Delta t/\tau)} \tag{11}$$

$$= e^{(k\ln m)} = m^k. \tag{12}$$

We can thus identify a relation between the branching parameter $m$ and the intrinsic timescale $\tau$ (or, more precisely, the autocorrelation time) via the time discretization $\Delta t$:

$$\tau = -\Delta t / \ln(m). \tag{13}$$

It is important to note that $\tau$ is an actual physical observable, whereas $m$ offers an interpretation of how the intrinsic timescales are generated—it sets the causal relation between two consecutive generations of activity. Whereas $m$ depends on how we chose the bin size of each time step $\Delta t$, the intrinsic timescale $\tau$ is independent of bin size.

## 4.2 Subsampling invariant estimation of the intrinsic timescale by multi-step regression

Subsampling describes the typical experimental constraint that often one can only observe a small fraction of the full system [5, 39, 40]. Given the full activity $A_t$, we denote the activity that is recorded under subsampling with $a_t$. We describe the amount of subsampling (the fraction of the system that is observed) through the sampling probability $\alpha$, where $\alpha = 1$ recovers the case of the fully sampled system.

It can be shown that subsampling causes a bias $b$ that only affects the amplitude of the autocorrelation function—but not the intrinsic timescale that characterizes the decay [5]. This is illustrated in Fig 3. By fitting the exponential and the amplitude, the subsampling problem boils down to an additional free parameter in the least-square fit of the correlation coefficients:

$$r_k = b\,m^k = b\,e^{-k\Delta t/\tau} \text{ with } b = \alpha^2 \frac{\text{Var}[A_t]}{\text{Var}[a_t]}, \tag{14}$$

where $a_t$ is the (recorded) activity under subsampling and $A_t$ is the (unknown) activity that would hypothetically be observed under full sampling. As we see above (Eq (14), Fig 3) the intrinsic timescale $\tau$ is independent of the sampling fraction $\alpha$. In general, when measuring autocorrelations, Eq (10), by definition $r_0 = 1$. Under subsampling however, the amplitude for $r_{k \geq 1}$ decreases as fewer and fewer units of the system are observed. This can cause a severe underestimation in the single regression approach, Eq (3).

In order to formalize the estimation of correlation coefficients $r_k$ forsubsampled activity, let us denote the set of all activity observations with $x = \{a_t\}$ and the observations $k$ time steps later with $y = \{a_{t+k}\}$. If $T$ is the total length of the recording, then we have $T - k$ discretized

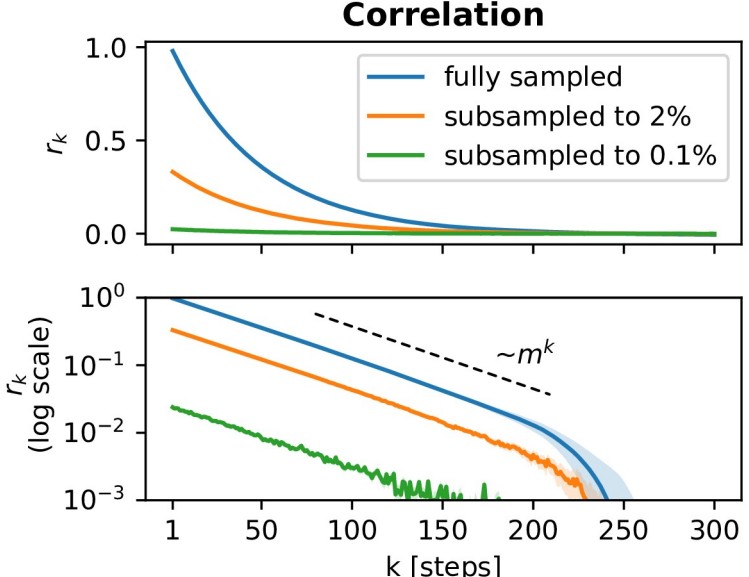

**Fig 3. The amplitude of correlation coefficients decreases under subsampling, whereas the intrinsic timescale $\tau$ and the branching parameter $m$ (characterized by the slope of the $r_k$ on a logarithmic scale) are invariant.** Coefficients were determined by the toolbox for a fully sampled and binomially subsampled branching processes [19].

time steps to work with. Then

$$r_k = \frac{\text{Cov}[x,y]}{\text{Var}[x]} = \frac{\sum_{t=1}^{T-k}(x_t - \langle x \rangle)(y_t - \langle y \rangle)}{\sum_{t=1}^{T-k}(x_t - \langle x \rangle)^2} , \tag{15}$$

where we approximate the expectation values $\langle x \rangle$ and $\langle y \rangle$ using

$$\bar{x} = \frac{1}{T-k}\sum_{t=1}^{T-k} a_t \text{ and } \bar{y} = \frac{1}{T-k}\sum_{t=1}^{T-k} a_{t+k} .$$

In other words, $\bar{x}$ is the mean of the observed time series and $\bar{y}$ is the mean of the *shifted* time series.

## 4.3 Different methods to estimate correlation coefficients

The drawback of the naive implementation, Eq (11), is that it is biased if $T$ is rather short—which is often the case if the recording time was limited (for a recent discussion of this topic see also Ref. [24]). In the case of short recordings, $\bar{x}$ and $\bar{y}$ are biased estimators of the expectation values $\langle a_t \rangle$ and $\langle a_{t+k} \rangle$. However, we can compensate the bias by combining multiple short recordings, if available.

In practice, multiple recordings are often available: If individual recordings are repeated several times under the same conditions, we refer to these repetitions as *trials*. One typically assumes that across these trials, the expected value of activity is stationary. However, this is not necessarily the case because trial-to-trial variability might be systematic. Since this assumption has to be justified case-by-case, the toolbox offers two methods to calculate the correlation coefficients: the `trialseparated` and `stationarymean` method.

**4.3.1 Trialseparated.** The `trialseparated` method makes less assumptions about the data than the `stationarymean` method. Each trial provides a separate estimate of the

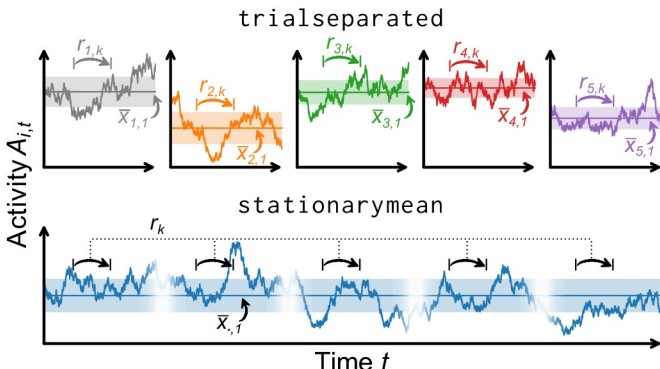

**Fig 4. Illustration of the two methods for determining the correlation coefficients $r_k$ from spiking activity $A_t$.**
Both methods assume a trial structure of the data (discontinuous time series)**Top**: The `trialseparated`
method calculates one set of correlation coefficients $r_{i,k}$ for every trial $i$ (via linear regression).**Bottom**: The
`stationarymean` method combines the information of all trials to perform the linear regression on a single, but
larger pool of data. This gives an estimate of $r_k$ that is bias corrected for short trial lengths.

correlation coefficients $r_{i,k}$. Let us again denote the observations before (after) the time lag
with $x_i$ ($y_i$), where index $i$ denotes the $i$-th out of $N$ total trials. All trials share the same number
of time steps $T$. We can apply Eq (15) to each trial separately and thereafter average over the
per-trial result:

$$r_k = \frac{1}{N}\sum_{i=1}^{N}\left[\frac{\sum_{t=1}^{T-k}(x_{i,t}-\bar{x}_{i,k})(y_{i,t}-\bar{y}_{i,k})}{\sum_{t=1}^{T-k}(x_{i,t}-\bar{x}_{i,k})^2}\right] = \frac{1}{N}\sum_{i=1}^{N}r_{i,k} \tag{16}$$

with

$$\bar{x}_{i,k} = \frac{1}{T-k}\sum_{t=1}^{T-k}a_{i,t} \quad \text{and} \quad \bar{y}_{i,k} = \frac{1}{T-k}\sum_{t=1}^{T-k}a_{i,t+k}.$$

As the expected activity $\langle a_t \rangle$ is estimated within each trial separately, this method is robust
against a change in the activity from trial to trial. On the other hand, the `trialseparated`
method suffers from short trial lengths when $\bar{x}_{i,k}$ and $\bar{y}_{i,k}$ become biased estimates for the
activity.

**4.3.2 Stationarymean.** The `stationarymean` method assumes the activity to be sta-
tionary across trials: Now, the expected activity $\langle a_t \rangle$ is estimated by $\bar{x}_{\cdot,k}$ and $\bar{y}_{\cdot,k}$ that use the full
pool of recordings (containing all trials):

$$r_k = \frac{\sum_{i=1}^{N}\left[\frac{1}{T-k}\sum_{t=1}^{T-k}(x_{i,t}-\bar{x}_{\cdot,k})(y_{i,t}-\bar{y}_{\cdot,k})\right]}{\sum_{i=1}^{N}\frac{1}{T}\sum_{t=1}^{T}(x_{i,t}-\bar{x}_{\cdot,k})^2} \tag{17}$$

with

$$\bar{x}_{\cdot,k} = \frac{1}{N(T-k)}\sum_{i=1}^{N}\sum_{t=1}^{T-k}a_{i,t} \quad \text{and} \quad \bar{y}_{\cdot,k} = \frac{1}{N(T-k)}\sum_{i=1}^{N}\sum_{t=1}^{T-k}a_{i,t+k}.$$

The two methods are illustrated in Fig 4 and the impact of the trial length on the estimated
autocorrelation time is shown in Fig 5. For short trials (red shaded area), the `stationary-`
`mean` provides precise estimates—already for time series that are only on the order of ten

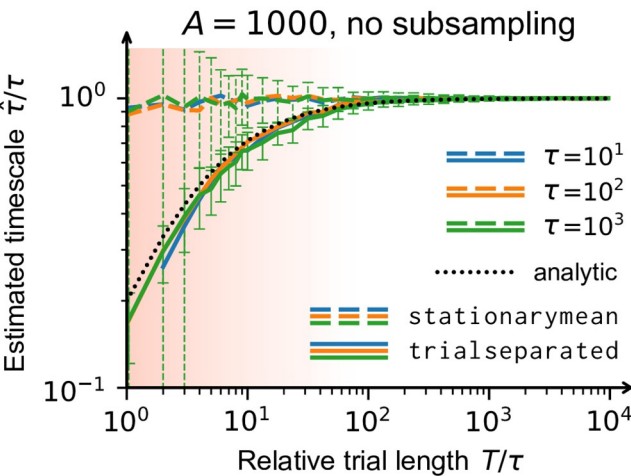

**Fig 5. Independent of subsampling, correlation coefficients can be biased if trials are short.** As a function of trial length, the autocorrelation time that was estimated by the toolbox ($\hat{\tau}$) is compared with the known value of a stationary, fully sampled branching process ($\tau$). Each measurement featured 50 trials and was performed once with each method, `trialseparated` (**solid lines**) and `stationarymean` (**dashed lines**). For short time series (**red shaded area**), it is known analytically that the correlation coefficients are biased [23]. The bias propagates to the intrinsic timescale (**black dotted line**) and it is consistent with the timescale obtained from the `trialseparated` method. The `stationarymean` method can compensate the bias, if enough trials are available across which the activity is indeed stationary. However, the improvement to the estimates scales directly with the number of trials—the effective statistical information is increased with each trial. Error bars (for clarity only depicted for $\tau = 10^3$): standard deviation across 100 simulations. For more details, see appendix B.

times as long as the autocorrelation time itself. The `trialseparated` method, on the other hand, is biased for short trials but it makes less strict assumptions on the data. Thus, the `trialseparated` method should be used if one is confident that trial durations are long enough.

As a rule of thumb, if an *a priori* estimate of $\tau$ exists, we advise to use trials that are at least 10 times longer than that estimate. The longer, the better. As an example, to reliably detect $\hat{\tau} \approx$ 200 ms (for instance in prefrontal cortex), a time series of 2 s could suffice (when using the `stationarymean` method). Furthermore, as a consistency check, we recommend to compare estimates that derive from both methods.

### 4.4 Toolbox interface to estimate correlation coefficients

The correlation coefficients are calculated by calling the `coefficients()` function, with the `method` keyword.

```
# typical keyword arguments, steps from 1 to 500

rks = mre.coefficients(src,   method = 'stationarymean',
steps = (1, 500))

# create custom steps as a numpy array,

# here from 1 to 750 with increment 2

my steps = np.arange(1, 750, 2)
```

```
# specify the created steps,
# step size dt and unit of the time-step
rks = mre.coefficients(src, method = 'stationarymean',
steps = my steps, dt = 1, dtunit = 'bp steps')
```

From the code example above, it is clear that one has to choose for which $k$-values the coefficients are calculated. This choice needs to reflect the data: the chosen steps determine the range that can be fitted. If not enough steps are included, the tail of the exponential is overlooked, whereas if too many steps are included, fluctuations may cause overfitting. A future version of the toolbox will give a recommendation, for now it is implemented as a console warning.

The $k$-values can be specified with the steps argument, by either specifying an upper and lower threshold or by explicitly passing an array of desired values. In order to give the $r_k$ physical meaning, the function also takes the time bin size $\Delta t$ (corresponding to the step size $k$) and the time unit as arguments: dt and dtunit, respectively. Those properties become part of the returned data structure CoefficientResult, so that the subsequent fit- and plot-routines can use them.

## 4.5 Toolbox data structure

Recordings are often repeated with similar conditions to create a set of trials. We took this into account and built the toolbox on the assumption that we always have a trial structure, even if there is only a single recording.

The trial structure is incorporated in a two dimensional NumPy array [20–22], where the first index ($i$) labels the trial. The second index ($t$) specifies the time step of the trials activity recording $A_{i,t}$, where time is discretized and each time step has size $\Delta t$. All trials must have the same length and the same $\Delta t$ (or in other words, should be recorded with the same sampling rate).

Because all further processing steps rely on this particular format, we provide the input_handler() that attempts convert data structures into the right one. The input_handler() works with nested lists, NumPy arrays or strings containing file paths. Wildcards in the file path will be expanded and all matching files are imported. If a file has multiple columns, each column is taken to be a trial. To select which of the columns to import, specify for example usecols=(0,1,2) which would import the first three columns.

## 4.6 Error estimation

The toolbox provides confidence intervals based on bootstrap resampling [41]. Resampling usually requires the original data to be cut into chunks (bins) that are recombined (drawing with replacement) to create new realizations, the so called bootstrap samples. Because the toolbox works on the trial structure, the input data usually does not need to be modified: each trial becomes a bin that can then be drawn with replacement to contribute to the bootstrap sample. While this is a good choice if sufficient ($\sim 100$) trials are provided, using trials directly for resampling means that no error estimates are possible with a single trial. If no trial structure is available, such as for resting-state data, an easy workaround is to manually cut long time

series into shorter chunks to artificially create the trial structure [19]. The error estimation via bootstrapping is implemented in the `coefficients()`, `fit()` and `full_analysis()` functions. All three take the `numboot` argument to specify how many bootstrap samples are created.

### 4.7 Getting help

Please visit the project on GitHub [42] and see our growing online documentation [43]. You can also call `help()` on every part of the toolbox:

```
# as an example, create variables.
bp = mre.simulate branching(m = 0.98, a = 10)
# try pressing tab e.g. after typing mre.c
rks = mre.coefficients(bp)
# help() prints the documentation,
# and works for variables and functions alike
help(rks)
help(mre.full analysis)
```

## 5 Discussion

Our toolbox reliably estimates the intrinsic timescale from vastly different time series, from electrophysiologal recordings to case numbers of epidemic spreading to any system that can be represented by an autoregressive process. Most importantly, it relies on the multi-step regression estimator so that unbiased timescales are found even for heavily subsampled systems [5].

In this work, we also took a careful look at how a limited duration of the recordings—a common problem in all data-driven approaches—can bias our estimator [23, 24]. With extensive numeric simulations we showed that the estimator is robust if conservatively formulated guidelines are followed. We can also bolster our previous claim [5] that the estimator is very data efficient. Moreover, short time series (trials) can be compensated by increasing the number of trials.

The toolbox thereby enables a systematic study of intrinsic timescales, which are important for a variety of questions in neuroscience [44]. Using the branching process as a simple model of neuronal activity, it is intuitive to think of the intrinsic timescale as the duration over which any perturbation reverberates (or persists) within the network [13, 45]. According to this intuition, different timescales should benefit different functional aspects of cortical networks [12, 46, 47].

Experimental evidence indeed shows different timescales for different cortical networks [5, 48]. It even suggests a temporal hierarchy of brain areas [1, 2, 49]; areas responsible for sensory integration feature short timescales, while areas responsible for higher-level cognitive processes feature longer timescales. For cognitive processes (for example during task-solving), the

intrinsic timescale was further linked to working memory. In particular, working memory might be implemented through neurons with long timescales [3, 4, 50].

In general, recordings could exhibit multiple timescales simultaneously [51–53]. This can be readily realized with the toolbox by using a custom fit function (e.g. a sum of exponential functions, see Sec. 3). However, it is important to be aware of the possible pitfalls of fitting elaborate functions to empirical data [53, 54]. In our experience, most recordings exhibit a single dominant timescale.

Lastly, it was theorized that biological recurrent networks can adapt their timescale in order to optimize their processing for a particular task [46, 55, 56]. For artificial recurrent networks, such a tuning capability was already shown to be attainable by operating around the critical point (of a dynamic second order phase transition) [15, 32, 47, 57]. For instance, reducing the distance to criticality increases the information storage in these networks [10, 12]. At the same time, the observed intrinsic timescale increases. It is plausible that the mechanisms of near-critical, artificial systems also apply to cortical networks [58–60]. This and other hypothesis can now be reliably tested with our toolbox and properly designed experiments [8]. For applications of our approach and the MR. Estimator toolbox see e.g. Refs. [48, 61, 62] and Ref. [7, 36, 63], respectively.

# 6 Appendix

## 6.1 A Real-world Example

**Listing 2**. Minimal script that shows how to prepare real-world data [30, 64], and produces Fig 2 from the main text. Characteristic for this dataset are theta oscillations (5–10 Hz) that carry over to the autocorrelation function. We first create a time series of activity by time-binning the spike times. Then, we create an artificial trial structure to demonstrate error estimation and apply the built-in fit functions. Last, we print the frequency $v = 6.13$ Hz of the theta oscillations as an example to show how to access the different parameters of the `complex` fit. The full script is available on GitHub [19], and for further details, also see the online documentation [43].

```
# helper function to convert a list of time stamps

# into a (binned) time series of activity

def bin_spike_times_unitless (spike_times, bin_size):
last_spike = spike_times [-1]

    num_bins = int (np. ceil (last_spike / bin_size))

    res = np. zeros (num_bins)

    for spike_time in spike_times:

        target_bin = int (np. floor (spike_time / bin_size))

        res [target_bin] = res [target_bin] + 1

    return res

# load the spiketimes

res = np. loadtxt ('./crcns/hc2/ec013.527/ec013.527.res.1')
```

```python
# the .res.x files contain the time-stamps of spikes
detected
# by electrode x sampled at 20 kHz, i.e. 0.05 ms per time
steps.
# we want 'spiking activity': spikes added up during a given
# time. usually, ~4ms time bins (windows) is a good first
guess
act = bin_spike_times_unitless (res, bin_size = 80)
# to get error estimates, we create 25 artifical trials by
# splitting the data. not recommended for non-stationary
data
triallen = int (np.floor (len (act)/25))
trials = np.zeros (shape = (25, triallen))
for i in range (0, 25):
    trials [i] = act [i * triallen: (i + 1) * triallen]
# now we could run the analysis and will get error estimates
# out = mre.full_analysis (trials, dt = 4, dtunit = 'ms',
kmax = 800,
# method = 'trialseparated')
# however, in this dataset we will find theta oscillations.
# let's try the other fit functions, too.
out = mre.full_analysis (trials, dt = 4, dtunit = 'ms',
kmax = 800, method = 'trialseparated', fitfuncs = ['exponen-
tial', 'exponential_offset', 'complex'], targetdir = './',
saveoverview = True)
# by assigning the result of mre.full_analysis (...) to a
# variable, we can use fit results for further processing:
# the oscillation frequency nu is fitted by the complex fit
# function as the 7th parameter (see online documentation).
# it is in units of 1/ dtunit and we used 'ms'.
print (f "theta frequency: {out.fits [2]. popt [6] * 1000}
[Hz]")
```

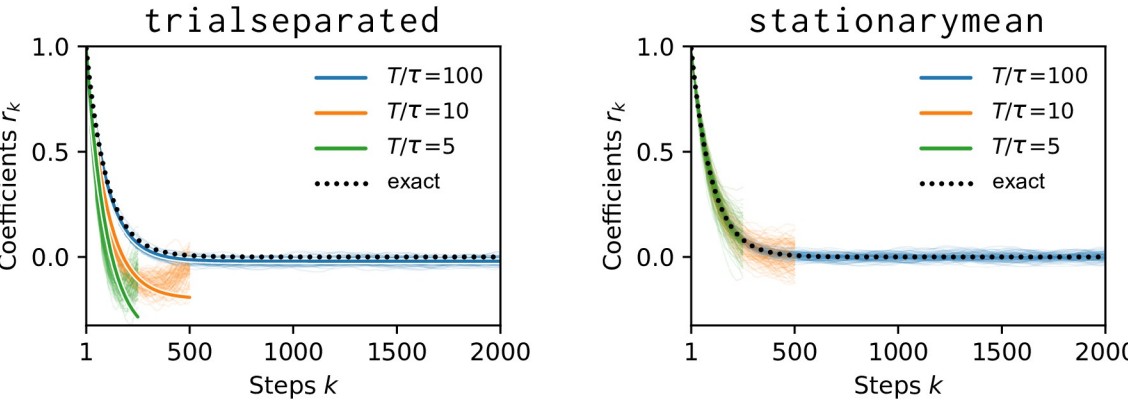

**Fig 6. Correlation coefficients $r_k$ for $\tau = 10^2$ (orange in Fig 5).** Individual background lines stem from the 100 independent repetitions.**Left**: Coefficients are shifted and skewed for short trial length $T/\tau$ when using the `trialseparated` method. The solid foreground lines are obtained from Eq. 4.07 of [23]. **Right**: With 50 trials and the `stationarymean` method, even very short (green) time series yield unbiased coefficients and, ultimately, precise estimates of the intrinsic timescale.

The spiking data from rats were recorded by Mizuseki et al. [30] with experimental protocols approved by the Institutional Animal Care and Use Committee of Rutgers University. The data were obtained from theNSF-founded CRCNS data sharing website [64].

## 6.2 B short trials cause bias

The data shown in Fig 5 was created with the `simulation_branching()` function included in the toolbox. Every measurement was repeated 100 times, featured 50 trials, target activity 1000 and no subsampling (the bias investigated here is independent from subsampling). The colored lines correspond to the median across 100 independent simulations. Error estimates were calculated but not plotted for clarity—in the red shaded area of Fig 5, the very short trials lead to low statistics (and large error bars). Error bars represent the standard deviation across the 100 simulations. The included steps $k$ covered $[1 : 20\tau]$, if available, which corresponds to the fit range of the exponential with offset.

To further illustrate the bias we observed in Fig 5, we plot the correlation coefficients $r_k$ that were found by the toolbox with the two different methods in Fig 6. When trials are short, the coefficients found by the `trialseparated` method are offset and skewed. The `stationarymean` method finds the correct coefficients because the estimation could profit from the trial structure. Since neither the true timescale nor the stationarity assumption are known in experiments, we suggest to compare results from both methods: if they agree, this is a good indication that the trials are long enough.

The black dashed line in Fig 5 is derived from the analytic solution Eq. 4.07 in Ref. [23] that gives the expectation value of the biased correlation coefficient in dependence of the trial length $T$. For simplicity, we focus on the leading-order estimated branching paramter $\hat{m}$ via the one-step autocorrelation function. Starting from Eq. 4.07 in Ref. [23],

$$\hat{m} \approx C(A_{t+1}, A_t) = r_1$$

$$\approx m^1 - \frac{1}{T}[(1+m)(1) + 2m^1] + O\left(\frac{1}{T^2}\right) \tag{18}$$

$$\approx m\left(1 - \frac{1}{T}\left[3 + \frac{1}{m}\right]\right) \tag{19}$$

cf. Eqs (4) and (11). Inserted into Eq (13) $\hat{\tau} = -\Delta t / \ln(\hat{m})$ and with $m = \exp(-\Delta t/\tau)$, we find

$$\hat{\tau} \approx \frac{-\Delta t}{\ln(m) + \ln\left(1 - \frac{1}{T}\left[3 + \frac{1}{m}\right]\right)} \tag{20}$$

$$\approx \frac{-\Delta t}{\ln(m) - \frac{1}{T}\left[3 + \frac{1}{m}\right]} \tag{21}$$

$$= \frac{-\Delta t}{\frac{-\Delta t}{\tau} - \frac{1}{T}\left[3 + e^{\Delta t/\tau}\right]} \tag{22}$$

$$= \frac{\tau}{1 + \frac{\tau}{T\Delta t}\left[3 + e^{\Delta t/\tau}\right]} . \tag{23}$$

For sufficiently large $\tau > \Delta t$, we obtain to leading order

$$\frac{\hat{\tau}}{\tau} \approx \frac{1}{1 + \frac{\tau}{T}\frac{4}{\Delta t}} . \tag{24}$$

For Fig 5—where $\Delta t = 1$, $x = T/\tau$ and $y = \frac{\hat{\tau}}{\tau}$—this means that

$$y = 1/(1 + 4/x) . \tag{25}$$

## Acknowledgments

We thank Leandro Fosque, Gerardo Ortiz and John Beggs as well as Danylo Ulianych and Michael Denker for constructive discussion and helpful comments. We are grateful for careful proofreading and input from Jorge de Heuvel and Christina Stier.

## Author Contributions

**Conceptualization:** F. P. Spitzner, J. Dehning, J. Wilting, J. Zierenberg, V. Priesemann.

**Data curation:** F. P. Spitzner, J. Dehning.

**Formal analysis:** F. P. Spitzner, J. Dehning.

**Investigation:** F. P. Spitzner.

**Methodology:** F. P. Spitzner, J. Wilting, J. Zierenberg, V. Priesemann.

**Project administration:** V. Priesemann.

**Software:** F. P. Spitzner, J. Dehning.

**Supervision:** V. Priesemann.

**Validation:** F. P. Spitzner, J. Dehning.

**Visualization:** F. P. Spitzner.

**Writing – original draft:** F. P. Spitzner, J. Dehning, J. Zierenberg.

**Writing – review & editing:** F. P. Spitzner, J. Dehning, J. Wilting, A. Hagemann, J. P. Neto, J. Zierenberg, V. Priesemann.

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
