## [Decision Letter · Decision Letter 0]

14 Dec 2020

PONE-D-20-31432

MR. Estimator, a toolbox to determine intrinsic timescales from subsampled spiking activity

PLOS ONE

Dear Dr. Spitzner,

Thank you for submitting your manuscript to PLOS ONE. After careful consideration, we feel that it has merit but does not fully meet PLOS ONE’s publication criteria as it currently stands. Therefore, we invite you to submit a revised version of the manuscript that addresses the points raised during the review process.

We look forward to receiving your revised manuscript.

Kind regards,

Michal Zochowski, Ph.D.

Academic Editor

PLOS ONE

Reviewers' comments:

Reviewer's Responses to Questions

**Comments to the Author**

1. Is the manuscript technically sound, and do the data support the conclusions?

Reviewer #1: Yes

2. Has the statistical analysis been performed appropriately and rigorously? 

Reviewer #1: Yes

3. Have the authors made all data underlying the findings in their manuscript fully available?

Reviewer #1: Yes

4. Is the manuscript presented in an intelligible fashion and written in standard English?

Reviewer #1: Yes

5. Review Comments to the Author

Reviewer #1: In this manuscript, Spitzner et al. introduce a Python package to extract intrinsic timescales from data that is well described by a subsampled AR process of first order. To this end, the package relies on the multiple-step regression method developed in (Wilting & Priesemann 2018). Beyond introducing the package and recapitulating the method, the authors investigate the bias due to short trials and derive an analytical approximation thereof. Given the broad applicability of AR processes and the ubiquity of the subsampling problem, it is certainly a valuable contribution - with potential impact across multiple disciplines - to provide an easily usable, well tested, and open source reference implementation.

The manuscript is well written and on point. Maybe more importantly, the user-level documentation of the package itself is excellent. After reading the paper and small parts of the documentation, it was straightforward to install the package and to use it on custom problems. On these problems, the package yielded the correct results and it seemed efficient although I did not perform any rigorous testing / profiling. Lastly, the code is well structured and documented.

I have only a few minor comments / remarks:

1. I think it would be good to include a 'real-world' use case to demonstrate the utility of the package.

2. What is the difference between the multi-step regression and an estimate of the normalized connected correlation function? Below Eq. (3) both approaches are called similar, below Eq. (5) it is stated that they are equivalent in the stationary case, but their difference is not discussed as far as I can see.

3. The role of stationarity in section 4.1 is not entirely clear to me. I think Eqs. (5-12) all assume stationarity. If this is correct I suggest to state it explicitly. On first reading, the "special case" seemed to apply only to Eq. (5) but not to the remainder of the section.

4. Please define both "intrinsic timescale" and "autocorrelation time" in the introduction to clarify in which context they are not synonymous.

5. Would it be sensible to use a logarithmic y-axis in Fig. 2 to visualize the invariance of the timescale?

6. The title of section 4.1 seems at mismatch with the content. Furthermore, this section is very dense and it might be worthwhile to expand it slightly and motivate the various manipulations a bit more.

7. Section 3 is not included in the overview at the end of the introduction.

8. Is there a particular reason why \\sum_i^N is used instead of \\sum_{i=1}^N, e.g. in Eq. (16)?

9. Figure 1 looks corrupted (black background, labels almost invisible) in the PDF but not when downloaded as a tiff file.

6. PLOS authors have the option to publish the peer review history of their article (what does this mean?). If published, this will include your full peer review and any attached files.

Reviewer #1: No

---

## [Author Response · Author response to Decision Letter 0]

25 Feb 2021

Dear Editors,

Dear Reviewer,

We thank the reviewer for the thoughtful comments and positive feedback. We made changes and additions according to the suggestions, and now include a real-world example in the appendix.

We want to address the suggestions point by point, in the attached response letter.

---

## [Editor Report · Decision Letter 1]

19 Mar 2021

MR. Estimator, a toolbox to determine intrinsic timescales from subsampled spiking activity

PONE-D-20-31432R1

Dear Dr. Spitzner,

We’re pleased to inform you that your manuscript has been judged scientifically suitable for publication and will be formally accepted for publication once it meets all outstanding technical requirements.

Kind regards,

Michal Zochowski, Ph.D.

Academic Editor

PLOS ONE

---

## [Editor Report · Acceptance letter]

20 Apr 2021

PONE-D-20-31432R1 

MR. Estimator, a toolbox to determine intrinsic timescales from subsampled spiking activity 

Dear Dr. Spitzner:

I'm pleased to inform you that your manuscript has been deemed suitable for publication in PLOS ONE. Congratulations! Your manuscript is now with our production department. 

Kind regards, 

on behalf of

Dr. Michal Zochowski 

Academic Editor

PLOS ONE